# Cases and distribution of visceral leishmaniasis in western São Paulo: A neglected disease in this region of Brazil

Regiane Soares Santana[1], Karina Briguenti Souza[1], Fernanda Lussari[1], Elivelton Silva Fonseca[1], Cristiane Oliveira Andrade[2], Marcia Mitiko Kaihara Meidas[3], Lourdes Aparecida Zampieri D'Andrea[4], Francisco Assis Silva[1], Edilson Ferreira Flores[5], Ivete Rocha Anjolete[6], Luiz Euribel Prestes-Carneiro[1] *

1 Department of Pós-Graduation, Environment and Regional Development Program, Oeste Paulista University, Presidente Prudente, São Paulo, Brazil, 2 Department of Control of Vectors, Teodoro Sampaio Municipality, Teodoro Sampaio, São Paulo, Brazil, 3 Bioclinic Clinical Laboratory, Teodoro Sampaio, São Paulo, Brazil, 4 Center for Biomedical Sciences and Regional Laboratory, Adolfo Lutz Institute, Presidente Prudente, São Paulo, Brazil, 5 Department of Statistics, School of Sciences and Technology, São Paulo State University, Presidente Prudente, São Paulo, Brazil, 6 Supervision and Control of Endemics, Presidente Prudente, São Paulo, Brazil

* luiz@unoeste.br

**Data Availability Statement:** All relevant data are within the paper and its Supporting Information files.

## Abstract

Visceral leishmaniasis (VL) is one of the most prevalent parasitic diseases worldwide. In 2019, 97% of the total numbers of cases in Latin America were reported in Brazil. In São Paulo state, currently 17.6% of infected individuals live in the western region. To study this neglected disease on a regional scale, we describe the spread of VL in 45 municipalities of the Regional Network for Health Assistance11(RNHA11). Environmental, human VL (HVL), and canine VL (CVL) cases, Human Development Index, and *Lutzomyia longipalpis* databases were obtained from public agencies. Global Moran's I index and local indicators of spatial association (LISA) statistics were used to identify spatial autocorrelation and to generate maps for the identification of VL clusters. On a local scale, we determined the spread of VL in the city of Teodoro Sampaio, part of the Pontal of Paranapanema. In Teodoro Sampaio, monthly peri-domicile sand fly collection; ELISA, IFAT and Rapid Test serological CVL; and ELISA HVL serum surveys were carried out. In RNHA11 from 2000 to 2018, *Lu. longipalpis* was found in 77.8%, CVL in 69%, and HVL in 42.2% of the 45 municipalities, and 537 individuals were notified with HVL. Dispersion occurred from the epicenter in the north to Teodoro Sampaio, in the south, where *Lu. longipalpis* and CVL were found in 2010, HVL in 2018, and critical hotspots of CVL were found in the periphery. Moran's Global Index showed a weak but statistically significant spatial autocorrelation related to cases of CVL (I = 0.2572), and 11 municipalities were identified as priority areas for implementing surveillance and control actions. In RNHA11, a complex array of socioeconomic and environmental factors may be fueling the epidemic and sustaining endemic transmission of VL, adding to the study of a neglected disease in a region of São Paulo, Brazil.

**Funding:** ESF reveiced funding from Fundação de Amparo à Pesquisa do Estado de São Paulo, Grant Number 2014/12494-0. The funders had no role in study design, data collection and analysis, decision to publish, or preparation of the manuscript.

**Competing interests:** The authors have declared that no competing interests exist.

## Author summary

Visceral leishmaniasis (VL) is considered the second most important disease caused by a protozoan worldwide. Currently, 97% of the cases in Latin America are in Brazil. In São Paulo state, since 1997, the disease has been found in an increasing number of municipalities, mainly in the western region. The reasons why VL is spreading in a crescent shape in this region is not well understood, however, socioeconomic and environmental risk factors that increase vulnerability to the disease may be involved. From 2000 to 2018, vectors of *Lutzomyia longipalpis* were found in 77.8%, canine VL in 69%, and human VL in 44.4% of the 45 municipalities of the western region. Dispersion occurred from the epicenter in the north and followed central and radial highways, reaching Teodoro Sampaio in the south. Among the 45 municipalities, 11 were identified as priority areas for implementing surveillance and control actions. Considered the last frontier and one of the poorest regions of São Paulo, our findings are relevant for planning interventions aimed at reducing the cases of VL in the region.

## Introduction

Visceral leishmaniasis (VL) is one of the most prevalent parasitic diseases worldwide. In 2017, 94% of new cases occurred in seven countries: Brazil, Ethiopia, India, Kenya, Somalia, South Sudan, and Sudan [1]. In 2019, 97% of the total numbers of cases in Latin America were reported by Brazil [2]. Currently, 25 of 27 Brazilian states notified autochthonous cases of canine visceral leishmaniasis (CVL) and 23 notified cases of human visceral leishmaniasis (HVL) [3].

In São Paulo, the richest and most populous state, *Lutzomyia longipalpis* sand flies were found in 1997, infected dogs in 1998, and infected humans in 1999. In 2020, 106 of 645 (16.4%) municipalities registered cases of HVL [4].

In recent years, from where and how VL reached the western region of São Paulo has been raised. Epidemiological studies demonstrated that the disease crossed the Bolivian border with Brazil and was found in the cities of Corumbá and Ladário, on the western side of the state of Mato Grosso do Sul (MS), expanded to its capital, Campo Grande, crossed the entire state and reached Três Lagoas, located on the western border of São Paulo. From MS to SP, the first diffusion occurred through the cities along with the construction of the Novoeste railway (1909–1952), the (BR-300) highway, completed in 1980, and the construction of a pipeline (1998) [5]. In São Paulo, CVL and HVL were first found in Araçatuba, a city harboring the railway, the highway, and the pipeline [6]. However, following the Candido Rondon highway (BR300), the disease crossed the state from the western region to the east toward the capital [6]. In the 45 municipalities of RNHA11 mesoregion, in the western region of São Paulo, in the border of MS, the first phlebotomine systematic collection was found in Dracena, a city located in the north and considered the epicenter [7]. In 2020, the western region harbored 537 of 3042 (17.6%) individuals infected with VL in São Paulo [4].

It has been hypothesized that in the western region, the presence of socioeconomic and environmental risk factors may favor the biological cycle of the disease, increasing the presence of *Leishmania (Leishmania) infantum chagasi* parasites, *Lu. longipalpis* vectors, and infected *Canis familiaris*, which in turn infect humans. Also, no study has been conducted on the dispersion of VL in the western region of São Paulo using an integrative approach so far. Our study is innovative because it allows the construction of a framework on two geographical scales: the regional scale combines public databases of the 45 municipalities of RNHA11

mesoregion; the local scale combines databases of Teodoro Sampaio, a representative city of RNHA11. To study the spread of this neglected disease in the 45 municipalities of RNHA11 on the regional scale, public databases on vectors, dogs, humans, environment, and socioeconomic risk factors were accessed. On a local scale, an entomological, canine, and human VL survey was conducted in Teodoro Sampaio, considered the capital of Pontal of Paranapanema.

## Methods

### Ethics statement

The project was approved by the ethics committee of Unoeste (number 4030). Canine samples were collected with the written informed consent of owners. Ethical approval was obtained from the Adolfo Lutz/Pasteur Institute Ethics Committee on the Use of Animals (number 02/2016). The study is for scientific research purposes and is in compliance with Federal Law no. 11794 of October 8, 2008, Decree no. 6899 of July 15, 2009, and in accordance with regulations outlined by The National Council for Animal Experimentation Control (Conselho Nacional para Controle da Experimentação Animal, CONCEA).

### Study design and setting

**Regional characteristics.** In 2020, São Paulo, the richest and most populous state of Brazil, with an estimated population of 44,639,899, accounts for 21.0% of the entire population of Brazil, estimated to be 213,930,425 inhabitants according to Brazilian Census (IBGE) [8] (Fig 1). The state comprises 645 municipalities, bordering Minas Gerais (north and northeast), Rio de Janeiro (northeast), Paraná (south), Mato Grosso do Sul (MS) (west), and the Atlantic Ocean (east) (Fig 1A). It is divided geographically into 15 mesoregions and 18 Regional Networks for Health Assistance (RNHAs). The western region comprises 45 municipalities and an estimated population of 753,344 inhabitants (2018) and it is administered by RNHA11, located in Presidente Prudente, mesoregion 8. The region is characterized by municipalities with a population <20,000 [8]. Among the 45 municipalities that are part of RNHA11, a group of 23 municipalities is called Pontal of Paranapema region, because of its relationship with the Paranapanema river. The Pontal of Paranapanema covers an area of 18,392.16 $km^2$ and has an estimated population of 620,953 [8] (Fig 1A). In contrast to the dynamic economic situation in the rest of the state, it is considered one of the poorest regions of São Paulo state [9].

**Local characteristics.** Teodoro Sampaio (latitude 22°31′57″ S and longitude 52°10′03″ W) at an altitude of 321 meters above sea level has an estimated population of 23,019. It is a small-sized urban town located 651 km from the state capital, and in the middle of a State Park and a high concentration of rural settlements. The municipality covers an area of 1,556 $km^2$, currently the 8th largest municipality in São Paulo [8] (Fig 1A and 1Ba). The municipality has sandy soil and a typical tropical climate with dry winters and wet summers, with an average annual temperature of 26°C varying from 16°C to 33°C.

The canine population of Teodoro Sampaio was estimated to be 5,754 animals, with a ratio of 1 dog per 4 inhabitants as previously determined for the region [10,11]. Devil's Hill State Park (Parque Estadual Morro do Diabo) is located 6.3 km from the urban area of Teodoro Sampaio. Covering an area of 338.5 $km^2$, it is one of the largest stretches of Tropical Forest in São Paulo state (Fig 1Bb). In the 1,950s, the whole region of RNHA11 was covered by a Tropical Forest that was almost completely destroyed. The Tropical Forest in Pontal of Paranapanema was extensively destroyed, especially in the middle of the 20th century, replaced by agricultural activities and urbanization. More than 3,000 $km^2$ of dense vegetation was reduced to 370 $km^2$ in less than 20 years (12.3%), including the Devil's Hill State Park and other small fragments. Flowing along the border of the city, the Paranapanema River is one of the biggest

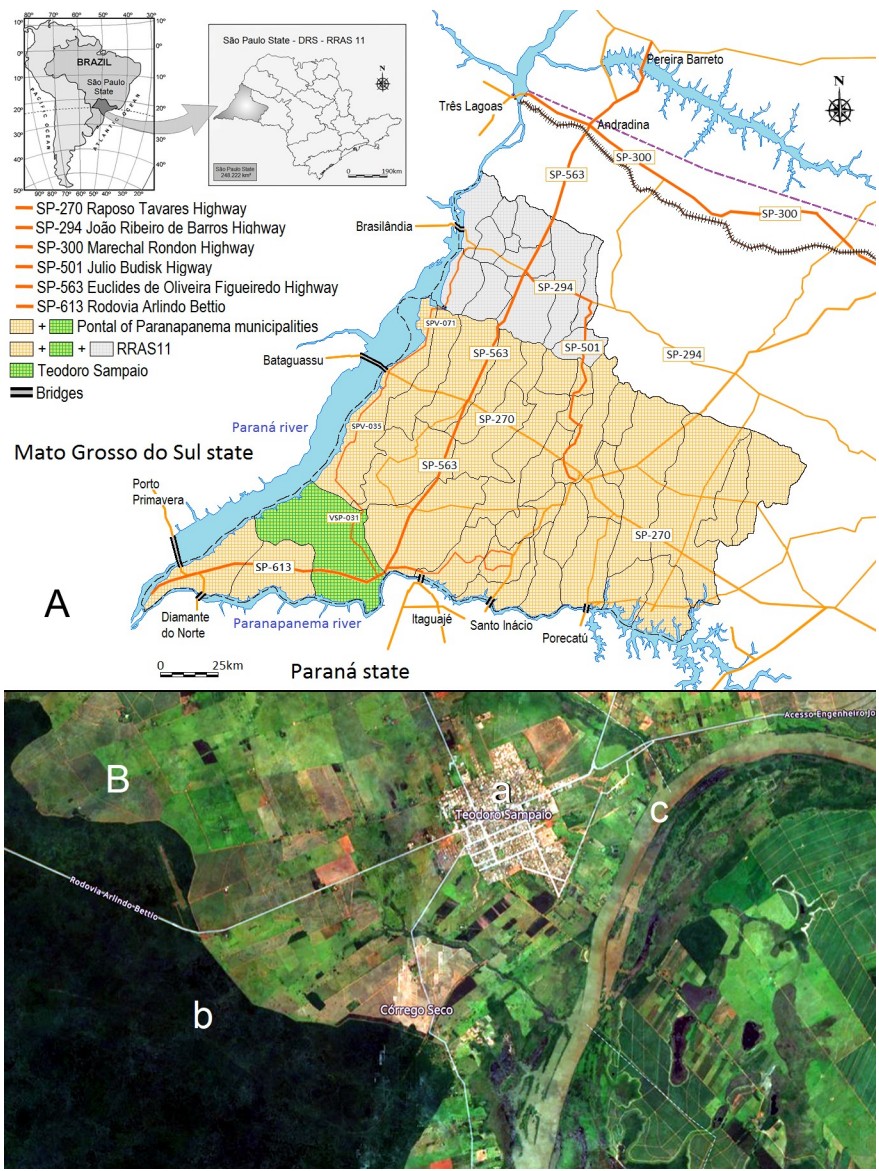

**Fig 1. Road network, rivers and artificial lakes, and location of Teodoro Sampaio in RNHA11 of São Paulo state.**
(A) The highway network between the municipalities of RNHA11 and Teodoro Sampaio. Artificial lakes and bridges on the Paraná River linking RNHA11 municipalities to well-known endemic regions in Mato Grosso do Sul state, and bridges over artificial lakes in the Paranapanema river linking regions with VL to VL-free areas in Paraná state. (B) OpenMapTiles map image of the municipality of Teodoro Sampaio: (a) the urban area of Teodoro Sampaio, (b) the Devil's Hill State Park, 6.3 km from the urban area, (c) the artificial lake and islands of the Paranapanema River, 0.9 km from the urban area. IBGE Cartographic Base Coordinate system: DATUM SIRGAS 2000. Website: https://www.ibge.gov.br/geociencias/organizacao-do-territorio/15774-malhas.html?=&t=downloads. OpenMapTiles map website. https://www.maptiler.com/maps/#hybrid//vector/11.88/-52.1721/-22.53428.

and most important rivers in the southeastern region of Brazil, and is a natural border between São Paulo and Paraná states (Fig 1Bc). It is 930 km long and since the 1,980s, it includes 11 large artificial lakes and an extensive flooded area, supporting hydroelectric plants and bridges. These bridges link dozens of small towns and villages connected by a large number of highways (Fig 1A).

## Potential dispersion routes of VL infection in RNHA11 mesoregion

RNHA11 is connected by two important radial highways: (1) the Marechal Candido Rondon (SP 300) linking two VL endemic regions: Três Lagoas, on the border of MS and Araçatuba-Bauru in the center-northeastern region of São Paulo; (2) the Euclides Figueiredo highway (SP 563) linking the VL endemic regions of Pereira Barreto, Ilha Solteira, and Andradina to Dracena, one of the most important cities of RNHA11, also with the highest prevalences in VL. Secondary highways also link the RNHA11 from north to south in direction to Paraná state, a region free of VL so far. Teodoro Sampaio is connected by a network of radial and secondary highways. One of the most important is the secondary vicinal highway (VSP-031; VSP-035) connecting the endemic regions of Presidente Epitácio and Panorama, bordering the Paraná River to Teodoro Sampaio (Fig 1A)

Data on *Lu. longipalpis*, canine visceral leishmaniasis (CVL) and HVL in the 45 municipalities of RHN11were obtained from archival databases from Brazilian public health agencies. *Lu longipalpis*: Supervision in Control of Epidemic (SUCEN), http://www.saude.sp.gov.br/sucen-superintendencia-de-controle-de-endemias/homepage/downloads/arquivos-leishmaniose-visceral; Canine visceral leishmaniasis: Center of Regional Laboratory of the Adolfo Lutz Institute of Presidente Prudente V (CRL-ALI-PPV), http://www.ial.sp.gov.br/ial/lab.-regionais/clr-v-presidente-prudente/contato; Human visceral leishmaniasis: Epidemiological Surveillance Center (CVE-SP), http://www.saude.sp.gov.br/cve-centro-de-vigilancia-epidemiologica-prof.-alexandre-vranjac/areas-de-vigilancia/doencas-de-transmissao-por-vetores-e-zoonoses/.

## Data on the presence of Lu. longipalpis in the municipalities of RNHA11

In RNHA11, data on phlebotomine sand flies identified in entomological collections were obtained from SUCEN in the period from 2000 to 2019. Entomologic surveys are proposed by the Manual of Surveillance and Control of Visceral Leishmaniasis of São Paulo (VLCPSP) [12], aiming to monitor the distribution of *Lu. longipalpis* in silent non-receptive, non-vulnerable municipalities, and silent non-receptive vulnerable municipalities as defined by the absence of confirmed autochthonous cases of HVL and CVL, without the presence of the chosen vector or not by estimated values of the distance to characterize the vulnerability. In these municipalities, an entomologic survey is recommended for the detection of the presence of *Lu. longipalpis* and to provide information on its distribution, allowing identification of the risk areas where control measures should be intensified [12].

## Canine and human visceral leishmaniasis and Center of Zoonosis Control in the municipalities of RNHA11mesoregion

According to VLCPSP guidelines [12], in São Paulo state, the municipalities are divided into two groups: silenced for VL, that is, without confirmation of autochthonous human and/or canine cases; and with transmission, that is, with confirmation of autochthonous human and/or canine cases. The investigation of a case of CVL may occur by a passive or house-to-house survey, conducted by the Center of Zoonosis Control (CZC) or by the Municipal Zoonosis Service (MZS) of each municipality. An active house-to-house search is conducted when *Lu. Longipalpis* is found in a particular area of the city or every time a dog is diagnosed with CVL; sequentially, all households within a radius of 200 m are visited and the dogs are tested for CVL. A passive survey is conducted when a dog's owner finds the symptoms described by the CZC or MZS as suspicious for CVL, as previously published [10]. Data on CVL, presence, and characteristics in each municipality under the CZC in the RNHA11 mesoregion were obtained from Adolfo Lutz Institute of Presidente Prudente [13]. A suspected case of HVL is confirmed

if it meets one of the following criteria: laboratory criteria, that is, identification of *Leishmania (Leishmania) infantum chagasi*, from a culture and/or inoculation in hamsters and/or by molecular techniques; presence of *Leishmania* spp. in the direct parasitological examination; presence of anti- *Leishmania (Leishmania) infantum chagasi* antibodies in serological tests and clinical-epidemiological criteria, that is, a patient from the area of transmission of VL, with suggestive clinical signs and without laboratory confirmation, provided that other diagnostic hypotheses are ruled out, that presents a favorable response to therapeutic treatment. Data on patients infected with HVL in RNHA11 were obtained from the Epidemiological Surveillance Center of São Paulo, which organizes and publishes data on recorded infectious diseases in São Paulo state annually [4].

## Mapping the cluster of identification of canine visceral leishmaniasis in RNHA11 mesoregion

In RNHA11 mesoregion, a Moran map was utilized to assess the formation of clusters of municipalities associated with the presence of CVL. Global spatial autocorrelation indicators, such as the Moran Index, present a single value as a measure of the spatial association for the data, which are presented by area, providing an index for the entire study region. However, it is often desirable to examine patterns on a more detailed scale to see if the process stationary hypothesis is verified locally, that is, if there is spatial dependence. Therefore, it is necessary to use spatial association indicators that can be associated with the different locations of a spatially distributed variable. This methodology uses the Local Moran Index (LISA) to find the spatial correlation of these areas. As it is a local indicator, there is a specific correlation value for each area, thus allowing the identification of clusters of areas and outliers [14,15]. Values vary from −1 to 1; values approaching zero indicate the absence of significant spatial autocorrelation of the values with their neighbors; values below 0.50 indicate a weak autocorrelation; values between 0.50 and 0.75 indicate medium autocorrelation; and values above 0.75 indicate strong autocorrelation. Positive values indicate a positive autocorrelation, where the value of what is being evaluated is similar to the values of its neighbors, and negative values indicate a negative autocorrelation [14–16].

## Mapping the Human Development Index in the RNHA11 mesoregion

The Human Development Index (HDI) was used as an indicator for understanding the socioeconomic profile of the municipalities of São Paulo state. The HDI is a well-known and comparable indicator worldwide, reproducing studies in other states and regions countrywide [17]. The HDI value of the 45 municipalities in the RHN11 was obtained in Fundação Sistema Estadual de Análises de Dados (SEADE), from 1999 to 2016, coded and classified by the denominated quintile as a reference parameter. Thematic maps were created for the HDI using ArcGis 10.7.1 software. The HDI was analyzed with support in a predictive method of interpolation surface generation: local polynomial interpolation (LPI). The Gaussian kernel method was used to generate the surface, classified according to the quintile. The gain in the LPI analysis in relation to the global polynomial interpolation presupposes it overlaps in several concentrations that a spatial representation may have, in our study, through a point in the centroid city of São Paulo state.

## Entomological survey in Teodoro Sampaio

In the urban area of Teodoro Sampaio, the phlebotomine sand fly entomological survey began in 2000, monthly, from February to October during the most favourable periods for detecting the presence of the vector. The survey ended in October 2010 when *Lu. longipalpis* was found in all areas. To standardize collections, the number (ranging from 1 to 40) and frequency (ranging from 1 to 39) of CDC light traps (Horst Ltd, São Paulo, Brazil) installed varied

according to the density and infestation of sand flies. The traps were placed approximately 1 m from the ground in places susceptible to the presence of phlebotomies and/or in the shelters of domestic animals, with an average temperature above 20˚C and relative humidity >70%. In each sector of the different areas, four areas with risk factors for the presence of phlebotomies were selected, including a large peri-domicile, and areas with abundant vegetation, accumulation of organic matter in the soil, and the presence of domestic animals from which the female sand fly can potentially obtain an infected blood meal. The traps were operated between 17:00 and 07:00 h. Monthly, four traps were installed for three nights in each sector. After capture, nylon-mesh cages were placed in plastic bags, tagged, and refrigerated (1˚C–7˚C) until identification was performed by the Regional Entomology Laboratories of SUCEN, Coordination of Disease Control, Secretary of State of Health, located in Presidente Prudente, São Paulo state. The sand flies were processed and identified using the taxonomic key of Galati [18].

## Canine visceral leishmaniasis survey in Teodoro Sampaio

The canine survey took place in four sectors of the urban area of Teodoro Sampaio, from January 2010 to December 2018, by a passive or house-to-house survey conducted by the Center of Zoonosis Control (CZC). We used recorded data of 3749 domiciled dogs of different ages, submitted to serum survey, using antibody tests to detect *Leishmania*, according to VLCPSP guidelines [12]. In Teodoro Sampaio, the Center of Regional Laboratory of the Adolfo Lutz Institute of Presidente Prudente (CRL-ALI-PPV) is responsible for the diagnostic tests for CVL. The VLCSP supervises the main actions to reduce morbidity and mortality, aimed at early diagnosis and treatment of human cases, vector control, identification, and euthanasia of seropositive domestic dogs. In 2009, the Ministry of Health began using enzyme-linked immunosorbent assays (ELISAs), produced by Bio-Manguinhos/Fiocruz, Ministry of Health, Rio de Janeiro, Brazil, for screening and confirmatory tests in CVL serological surveys. For ELISA-positive samples, a confirmatory test using IFAT, produced by Bio-Manguinhos/Fiocruz, was performed. From April 2012 to the present, the Brazilian Ministry of Health replaced the ELISA/IFAT tests with a new protocol using a dual-path platform CVL rapid test, produced by Bio-Manguinhos/Fiocruz, with ELISA-CVL as a confirmatory test. Animals with positive samples in the screening test (ELISA from January 2009 to March 2012 and the dual-path platform CVL rapid test from April 2012 until the present and a negative result in the confirmatory test (IFAT and ELISA, respectively) were re-sampled. Only concordant samples that were positive in the screening test and positive in the confirmatory test were considered seropositive [10]. Data on the presence and characteristics in each municipality under the CZC in the RNHA11 mesoregion were obtained from Adolfo Lutz Institute of Presidente Prudente [13].

## Human visceral leishmaniasis in Teodoro Sampaio

Data on human leishmaniasis was obtained from the Epidemiological Surveillance Municipality of Teodoro Sampaio. The sole case of HVL in Teodoro Sampaio was referred to the Regional Hospital of Presidente Prudente, a tertiary, university, public, and reference center for treatment of HVL in RNHA11 mesoregion. Confidentiality was assured. The case was confirmed in 2018 by clinical/epidemiologic criteria and laboratory diagnostics according to the VLCPSP guidelines [12].

## Human visceral leishmaniasis serological survey and laboratory diagnosis in Teodoro Sampaio

In July 2018, a serological survey was conducted on 159 individuals aged from 18 to 65 years from four different locations: a health care center, a Family Health Strategy Unit; the Social Assistance Reference Center; a State Technical School; and in individuals attending a private

clinical laboratory. Blood was collected blood collections tubes under vacuum with and without EDTA as an anticoagulant. Serum was separated by centrifugation and stored at −200˚C until use. The analysis was performed at the CRL-ALI-PPV using a qualitative test for the detection of antibodies against HVL by ELISA, according to the manufacturer's instructions (LEISHMANIA ELISA IgG+IgM; VIRCELL, Santa Fe, Granada, Spain). The test uses recombinant antigens capable of fixing specific antibodies to *Leishmania (Leishmania) infantum chagasi* with a sensitivity of 97% and specificity of 99%. Positive tests were confirmed by indirect immunofluorescence (Bio-Manguinhos/Fiocruz, Ministry of Health, Rio de Janeiro, Brazil). Hemograms and hepatic enzymes were determined because hepatic function may be affected and pancytopenia may be present in patients infected with VL. Hemograms were performed using a flow cytometer flux counter (Pentra 80, HoribaDiagnostics, Montpellier, France) and the differential leukocyte count was compared with direct microscopic observation of blood smears. Alanine aminotransferase (ALT), aspartate aminotransferase (AST), and albumin were assessed using automated systems according to the manufacturer's instructions; the normal ranges were 7–55 U/L, 8–43 U/L, and 3.5–5 g/dL, respectively. A total of 16,636 individuals were considered to determine the sample size. These samples were processed by Laboratório Bioclínico, Teodoro Sampaio, SP, Brazil.

## Kernel maps construction

Dogs that underwent the serological survey and those with positive serology were registered according to the location used by the municipal health surveillance service. The city was divided into four sectors, starting from the northeast of the urban area in a clockwise direction. These sectors, in turn, were divided into 432 blocks, arranged in a polygon shape, used to generate the Geographic Information System and served as a basis for indexing the attributes of the surveyed blocks. With the geometric points generated per block, it was possible to visualize the number of dogs with CVL in each block, the number of dogs in the census for each block, and the number of euthanized dogs per block. The number of positive dogs per block is not consistent with the number of euthanized dogs, because there were some cases in which euthanasia of dogs that tested positive was not performed. For the procedure that identified hot areas of CVL infection, we used the Kernel density maps, using the euthanized dogs as a population field to generate a continuous surface to the urban area of the Teodoro Sampaio. A raster surface was generated using the Gaussian equation, with a search ratio equivalent to a block (100 m) and a pixel size equivalent to a household (10 m). The coordinate system used as reference was WGS1984. Regarding the parameters for reading the data for analysis, the urban area of Teodoro Sampaio measures $4 \times 3$ km$^2$.

## Statistical analysis

The results are shown as means standard error of the mean (SEM) (for normally distributed variables). Dichotomous and nominal variables are expressed as frequencies and percentages. Pearson's correlation coefficient was used to describe the functional relationship between the number of cases of HVL per 10,000 inhabitants and the HDI of each municipality. Statistical analysis was performed using GraphPad Software (San Diego, CA, USA) and the Sigma-Stat program (Systat Software, Richmond, CA, USA). ArcGIS 10.7 and ArcGis Pro were used for data analysis and layout design.

## Results

### The presence of Lu. longipalpis in the municipalities of RNHA11 mesoregion

In São Paulo, *Lu. longipalpis* sand flies were found for the first time in Araçatuba, an area surrounding the western region in January 1997. From 1997 to July 2019, the vector was found in

203 of 645 (31.5%) of the municipalities. In RNHA11 mesoregion, *Lu. longipalpis* was detected for the first time in Dracena in 2003 in the north; the sand flies dispersed toward the south in the direction of the municipalities of Pontal of Paranapanema and Paraná state. In July 2019, *Lu. longipalpis* was found in 35 of 45 (77.8%) of the municipalities of RNHA11 mesoregion.

### Canine visceral leishmaniasis cases and the presence of the Center of Zoonosis Control in municipalities of RNHA11mesoregion

In RNHA11, CVL was described for the first time in Dracena in 2005. In May 2018, CVL transmission was present in 29 of 45 (64.4%) of municipalities of RNHA11. Among the municipalities, 13 of 45 (28.9%) do not have CZC, 4 of 45 (8.9%) have a partial physical and human structure of CZC, and 28 of 45 (62.2%) have a structured CZC (Fig 2).

### Spatial cluster analysis of CVL cases in RNHA11 mesoregion

The cases of CVL in the 45 municipalities of RNHA11 mesoregion during the period from 1999 to 2019 was totalled by the municipality. The Global Moran Index was applied and presented a weak spatial autocorrelation, but statistically significant (I = 0.2572). Applying the Local Moran Index Cluster Map (LISA) revealed that most municipalities (32 of 45; 71.1%) were non-significant, however, 13 of 45 (28.9%) showed significant spatial autocorrelation, distributed in the following quadrants: high-high with 5 municipalities Q (+/+), Presidente

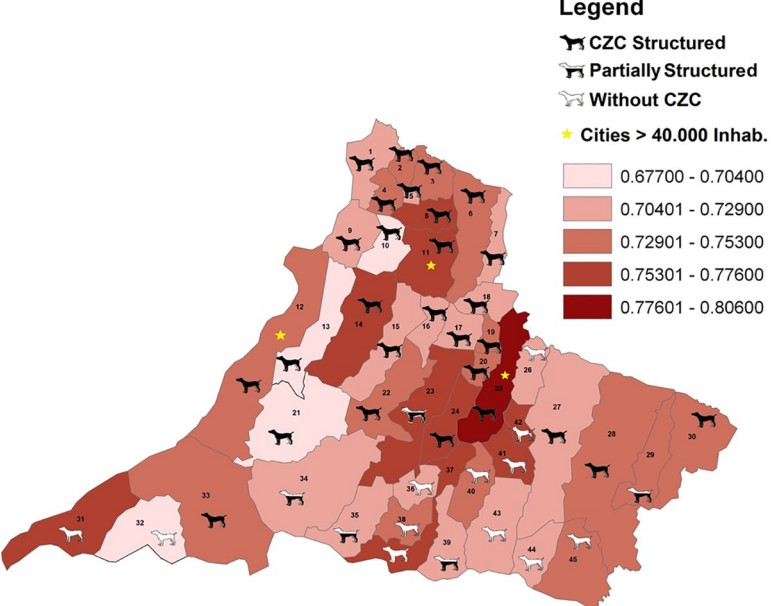

**Fig 2. Spatial distribution of the HDI, the presence of CZCs, and populations >40,000 inhabitants in municipalities of RNHA11 mesoregion.** Source: State System Data Analysis Foundation (SEADE, 2017). Base map: digital meshes of IBGE [8]. The municipalities are numbered as follows: 1, Paulicéia; 2, São João do Pau D'alho; 3, Monte Castelo; 4, Santa Mercedes; 5, Nova Guataporanga; 6, Junqueirópolis; 7, Irapuru; 8, Tupi Paulista; 9, Panorama; 10, Ouro Verde; 11, Dracena; 12, Presidente Epitácio; 13, Caiuá; 14, Presidente Venceslau; 15, Piquerobi; 16, Ribeirão dos Índios; 17, Emilianópolis;18, Flora Rica;19, Santo Expedito; 20, Alfredo Marcondes; 21, Marabá Paulista; 22, Santo Anastácio; 23, Presidente Bernardes; 24, Alvares Machado; 25, Presidente Prudente; 26, Caiabu; 27, Martinópolis; 28, Rancharia; 29, João Ramalho; 30, Quatá; 31, Rosana; 32, Euclides da Cunha Paulista; 33, Teodoro Sampaio; 34, Mirante do Paranapanema; 35, Sandovalina; 36, Tarabai; 37, Pirapozinho; 38, Estrela do Norte; 39, Narandiba; 40, Anhumas; 41, Regente Feijó; 42, Indiana; 43, Taciba; 44, Nantes; 45, Iepê. Map of the Municipalities: IBGE Cartographic Base Coordinate system: DATUM SIRGAS 2000. Website: https://www.ibge.gov.br/geociencias/organizacao-do-territorio/15774-malhas.html?=&t=downloads.

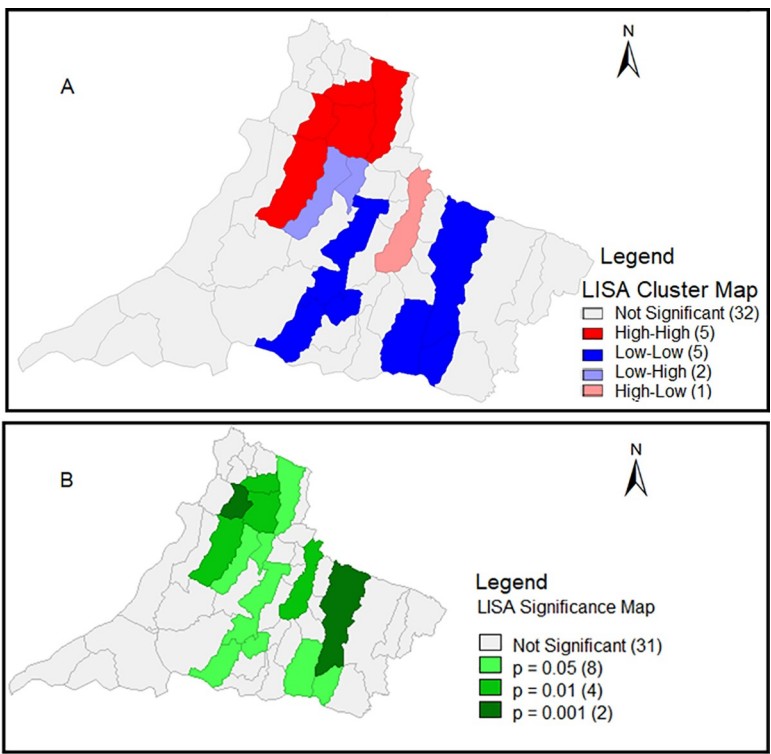

**Fig 3. Spatial distribution of high-risk clusters for CVL transmission, 2005–2018.** Among the 45 municipalities of RNHA11, 11 are classified as high-risk clusters. They are concentrated mainly in the north region, considered the epicenter of the disease. (A) Univariate LISA and (B) LISA significance. Map of the Municipalities: IBGE Cartographic Base Coordinate system: DATUM SIRGAS 2000. Website: https://www.ibge.gov.br/geociencias/organizacao-do-territorio/15774-malhas.html?=&t=downloads Coordinate system: DATUM SIRGAS 2000. Website: https://www.ibge.gov.br/geociencias/organizacao-do-territorio/15774-malhas.html?=&t=downloads. Map with Spatial Statistics Application GEODA: https://spatial.uchicago.edu/software.

Venceslau, Ouro Verde, Dracena, Tupi Paulista, and Junqueirópolis; low-low with 5 municipalities Q (−/−), Sandovalina, Tarabai, Martinópolis, Taciba, and Presidente Bernardes; low-high with 2 municipalities Q (−/+), Piquerobi and Ribeirão dos Índios; and in the high-low quadrant, only one municipality Q (+/−), Presidente Prudente. In the LISA map, we can see the p value of significant municipalities (Fig 3).

## Human visceral leishmaniasis in RNHA11 mesoregion

Fig 4A shows that over 14 years of follow-up (2005–2018), 19 of 45 (42.2%) municipalities in RNHA11 mesoregion referred patients infected with VL. In the northern area, all the municipalities' notified cases, most of them at increased rates (17–57 cases), and Dracena (dark red), considered to be the epicenter, showed the highest rates (58–176 cases). On Pontal of Paranapanema, 8 of 32 (25.0%) municipalities registered HVL, and dispersion on the regional scale occurred in three different ways: irradiation, contiguity, and jumps (when municipalities far from the endemic region register cases of CVL or HVL). In municipalities located along the Paraná River, only one did not register cases (Fig 4A). Fig 4B shows that cases were found first in Dracena in 2005, in subsequent years HVL spread to the neighboring municipalities and in 2010 reached the municipalities of Pontal of Paranapanema. Teodoro Sampaio, 133 km from the epicenter, was one of the last municipalities to register cases and HVL was detected in 2018.

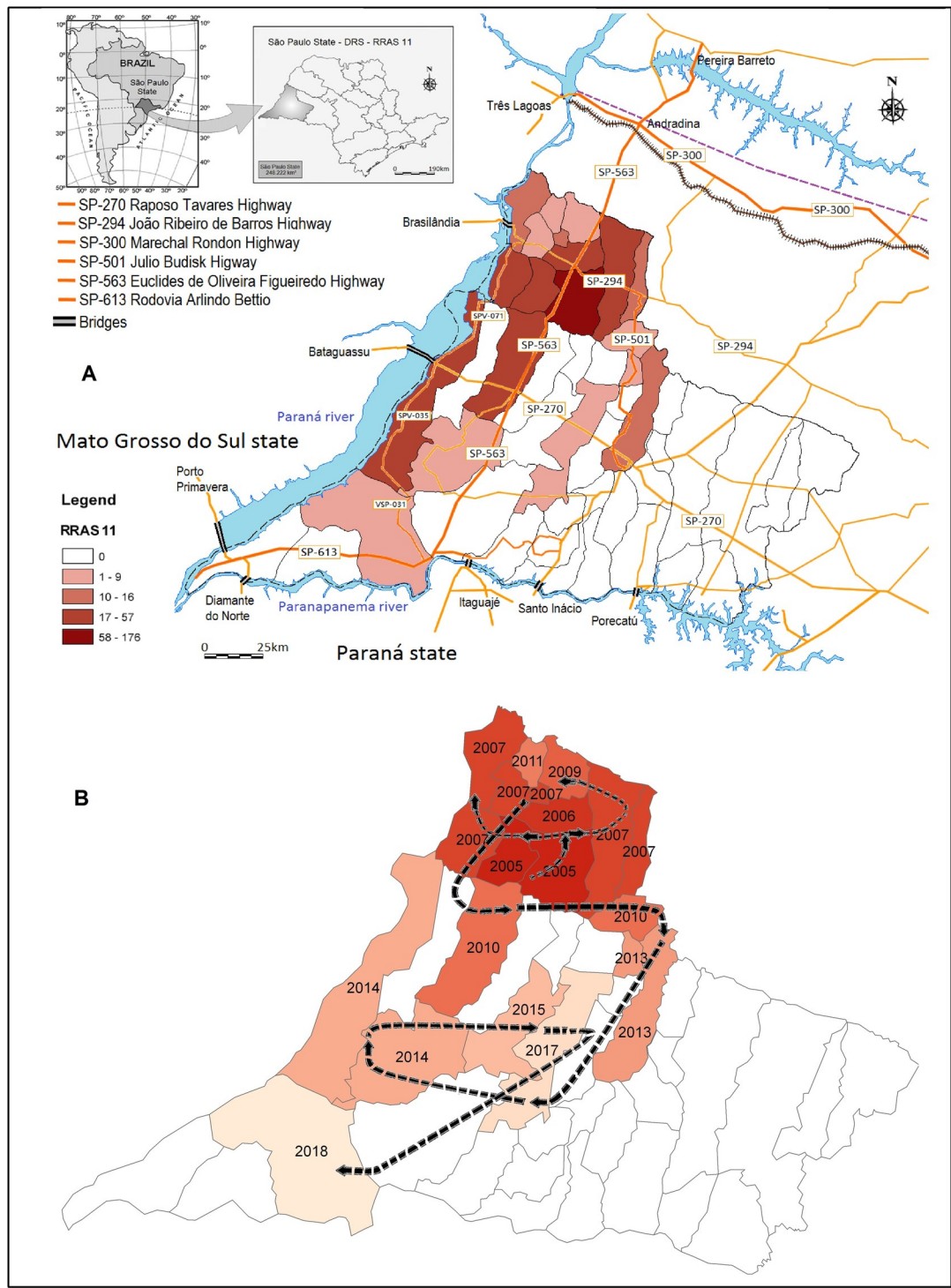

**Fig 4. Geographic distribution of human visceral leishmaniasis in the RNHA11 mesoregion between 2005 and 2018.** The colors represent the cumulative number of cases (A). Flow and spatial dispersion of human visceral leishmaniasis from 2005 to 2018 in the RNHA11 mesoregion, and the year in which the first case was detected in each municipality (B). IBGE Cartographic Base Coordinate system: DATUM SIRGAS 2000. Website: https://www.ibge.gov.br/geociencias/organizacao-do-territorio/15774-malhas.html?=&t=downloads.

## Paraná and Paranapanema Rivers, dams and artificial lakes in RNHA11 mesoregion and surrounding areas of Teodoro Sampaio

Figs 1A and 4A show that the municipalities in the RNHA11 mesoregion are divided from MS on the western side by the Paraná River, which flows along the border of five cities. On the south side, the region is divided from Paraná state by the Paranapanema River, which flows along the border of ten municipalities. Together, the rivers support seven large lakes and a flooded area of 6175 km$^2$ and seven dams and bridges (Fig 1A). These bridges link the VL endemic regions of Três Lagoas in MS and Andradina in São Paulo state to the areas of the western region as well as the municipalities of Pontal of Paranapanema to the VL-free areas of Paraná state. The urban area of Teodoro Sampaio (Fig 1Ba) is surrounded by the Devil's Hill State Park (Fig 1Bb), 6.3 km away, and linked by a highway with the administrative offices, museums, laboratories, and environmental educational areas of the park. The municipality of Teodoro Sampaio is served by both the Paraná and Paranapanema Rivers, which surround the urban area of the city at a distance of 1 km (Fig 1Bc). In the region, the average annual precipitation varies between 1200 and 1500 mm, and the average annual temperature remains slightly above 22˚C.

## The Human Development Index in the RNHA11mesoregion

In the municipalities in this study, the HDI varied from very low in 4 municipalities and low in 15 municipalities to high only in Presidente Prudente, the most important city in RNHA11 mesoregion. Of the areas with very low and low HDIs, 18 (40%) of the municipalities are located in the Pontal of Paranapanema region. Ribeirão Preto, connecting the Paraíba Valley to the Metropolitan region of São Paulo city, and from Campinas to Piracicaba presents an axis of great development, with the HDI varying from 0.790 to 0.826. The regions with the lowest HDI values are the Ribeira Valley, located in the south of the state, and Pontal of Paranapanema (our study area). In Teodoro Sampaio, rates vary from 0.678 to 0.715. In São Paulo state, the HDI is distributed by regions, and maps are available in Fundação Sistema Estadual de Análise de dados (SEADE), Teodoro Sampaio is surrounded by very poor municipalities, Euclides da Cunha Paulista, Marabá Paulista, and Caiuá, with rates between 0.6770 and 0.704 (Fig 2).

When the functional relationship between the number of cases of HVL per 10,000 inhabitants and the HDI of each municipality was calculated, the correlation (adjusted R value) had a value less than 10% and the *P* value was not significant, however, when municipalities that did not have cases of VL were removed and the simple linear regression technique was applied, the regression line showed a trend for lower values of VL cases per 10,000 inhabitants as the HDI increased (Fig 5).

## Entomological survey in the urban area of Teodoro Sampaio

Teodoro Sampaio covers two areas with regard to operational disease control programs (e.g. dengue fever surveillance and control, CVL and HVL surveillance and control). The two areas are divided into four sectors and the sectors are divided into blocks (Fig 6A and 6B). *Lu. longipalpis* were initially found in July 2010 in area 1, sectors 2 and 4 in blocks 146 and 298, and in October 2010 in area 1, sector 1, and block 75 (Table 1).

## Canine serological survey and spatial distribution of euthanized dogs in Teodoro Sampaio

In 2010, CVL was identified for the first time in the urban area of Teodoro Sampaio. From January 2010 to December 2018, 3749 dogs were selected for serology by the CZC and 298 animals

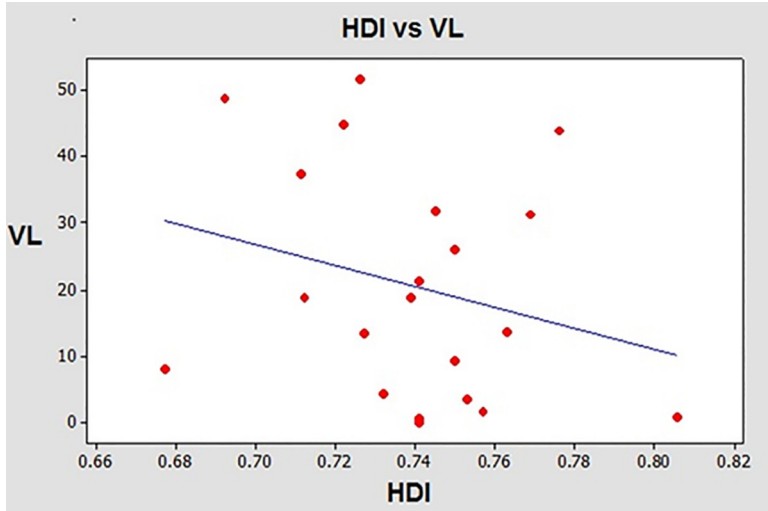

**Fig 5. Correlation between HDI and VL infection in municipalities in RNHA11 mesoregion.**

(7.9%) were positive for CVL. Fig 6A shows the kernel spatiotemporal distribution of euthanized dogs. Of the 298 dogs that were positive, 110 (36.9%) were euthanized according to the criteria of the Brazilian Ministry of Health. In 2010 and 2011, infected dogs were identified

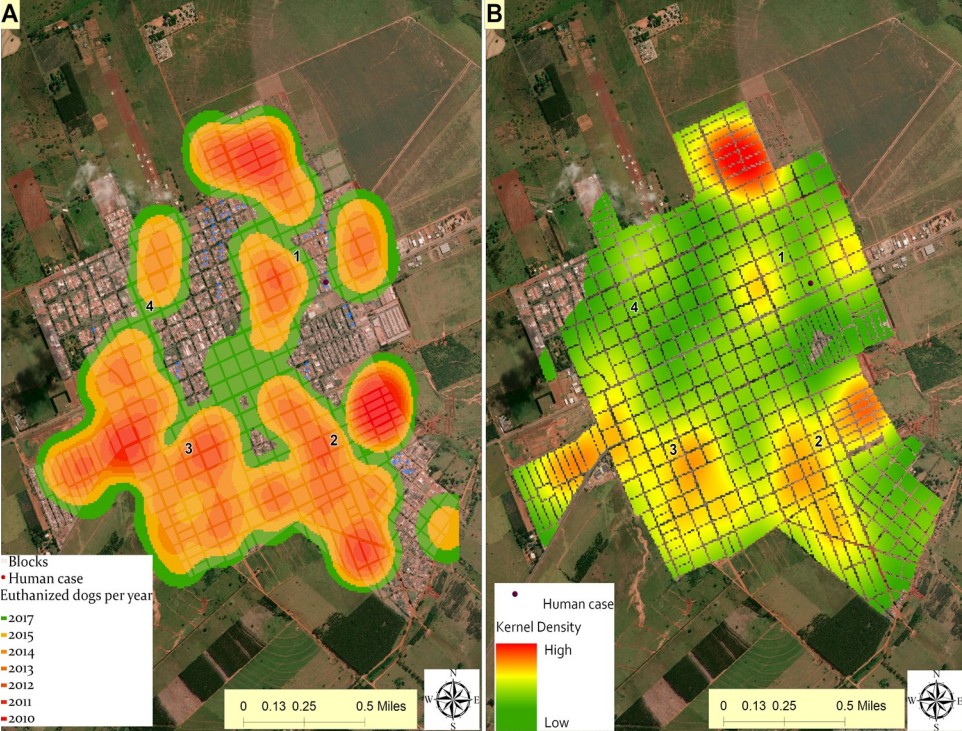

**Fig 6. Kernel of spatiotemporal distribution and density estimation of CVL seropositive euthanized dogs in Teodoro Sampaio, 2010–2018.** Kernel of the spatiotemporal distribution of CVL diagnosed and euthanized dogs (A). The hotspots represent areas where a higher density of seropositive dogs was observed, from 2010 to 2018 (B). In the map legend, the lighter colors represent the low density of CVL cases and the darker colors represent the presence of hotspots. Data on seropositive euthanized dogs are available as S1 Data file.

**Table 1. Spatiotemporal distribution of *Lu. longipalpis* sand flies in the urban area of Teodoro Sampaio, São Paulo, Brazil.**

| MUNICIPALITY OF TEODORO SAMPAIO | | | | | | |
|---|---|---|---|---|---|---|
| AREA | SECTOR | BLOCK | DATE | No. OF TRAPS | No. OF TIMES | SPECIES/NUMBER |
| 1 | 2 | 146 | 22 July 2010 | 9 | 9 | *Lu. longipalpis* 1 male |
| 1 | 4 | 298 | 27 July 2010 | 10 | 10 | *Lu. longipalpis* 1 male |
| 1 | 1 | 75 | 25 October 2010 | 14 | 14 | *Lu. longipalpis* 1 male |
| | | Total | | 33 | 33 | 3 males |

and euthanized in sectors 1, 2, and 3 (three hotspots; dark red color). In 2012–2013, infected dogs were found in the areas surrounding the sectors where dogs positive for CVL were found in 2010–2011 (ochre color). From 2014 to 2017, positive cases reached the entire urban area, including sector 4 (apple green and green colors), and in 2018 no euthanized dogs were registered. Fig 6B shows the hotspots of infected dogs euthanized. The periphery of sector 1 showed a higher number, demonstrated by critical points hotspots, whereas sector 4 showed the lowest number. These data allow us to consider that a stronger force of infection occurs in sector 1, the main hotspot. The first case of HVL in the city was located in this sector. Several dogs were not euthanized for different reasons.

## Cases of human visceral leishmaniasis in Teodoro Sampaio

Teodoro Sampaio, 133 km from the epicenter, was one of the last municipalities to register cases and HVL was detected in 2018. The first autochthonous case of HVL was identified in 2018 in the eastern area, area 1, on the outskirts of Teodoro Sampaio (Fig 6B), where a high number of cases of CVL were found throughout the period. The patient was a 48-year-old man who did not respond to treatment with meglumine antimoniate and was treated with liposomal amphotericin B (5 mg/kg/day) for 5 days with improvement in the symptoms and 50% reduction of splenomegaly.

## Human visceral leishmaniasis serological survey, laboratory findings and epidemiologic characteristics of the study population in Teodoro Sampaio

Between January and December 2018, 159 individuals living in Teodoro Sampaio were enrolled in the human visceral serological survey, and 2 of 159 (1.3%) resulted positive; 4 of 159 (2.5%) resulted indeterminate and 153/159 (96.2%) resulted negative. Hematologic abnormalities are among the most common manifestations in patients infected with VL in whom pancytopenia may occur (Table 2). From 159 individuals analyzed, mild anemia occurred in 5 of 159 (3.1%) individuals, leukopenia occurred in 3 of 159 (1.9%), and thrombocytopenia occurred in 4 of 159 (2.5%). The mean values±SEM, with 95% confidence intervals (CI) were as follows: red blood cells, 4.81±0.03 million/ cells/mcL (95% CI, 4.50–4.62); hemoglobin (g/dL), 13.80±0.10 (95% CI, 13.59–14.00); hematocrit (%), 40.33±0.27 (95% CI, 39.78–40.88); mean corpuscular volume (fL), 88.49±0.33 (95% CI, 87.83–89.15); mean corpuscular hemoglobin (pg), 31.50±1.25 (95% CI, 29.02–33.99); mean corpuscular hemoglobin concentration (g/dL), 34.20±0.06 (95% CI, 34.07–34.34); platelets ($\times 10^3$/mcL), 266.359±4.740 (95% CI, 256.997–275.720). When the main changes in white blood cells were analyzed in the whole population, 17 of 159 (10.7%) showed mild leucopenia, 13 of 159 (8.1%) had mild leukocytosis, 25 of 159 (15.7.2%) had neutropenia, 41 of 159 (25.8%) had eosinophilia, 18 of 159 (11.3%) had lymphocytosis, and 11 of 159 (6.9%) had monocytosis. Mild thrombocytopenia was present in 9 of 159 (5.6%) individuals.

**Table 2. Hematologic and biochemical features recorded in 159 individuals included in the human serological survey in Teodoro Sampaio, São Paulo state, Brazil, in July 2018.**

| PARAMETER | NUMBER (%) | MEAN±SD | 95% CI | NR |
|---|---|---|---|---|
| Erythrocytes (million/μL) | | | | |
| Men <4.5 | 3 (6.6) | 4.207±0.1677 | 3.790–4.623 | 5.00±0.5 |
| Women <3.8 | 3 (2.6) | 3.537±0.1050 | 3.276–3.798 | 4.3±0.5 |
| Hemoglobin (g/dL) | | | | |
| Men <13 | 1 (2.2) | 12.50 | | 15.0±2.0 |
| Women <12 | 9 | 11.48±0.4410 | 11.14–11.82 | 13.5±1.5 |
| Hematocrit (%) | | | | |
| Men <40 | 4 (8.8) | 38.58±0.8261 | 37.26–39.89 | 45±5 |
| Women <39 | 57 (50) | 36.90±1.632 | 36.47–37.33 | 41±5 |
| MCV <83 (fL) | 11 (6.9) | 78.84±2.599 | 77.09–80.58 | 92±9 |
| MCH <27 (pg) | 6 (3.7) | 25.70±0.8809 | 24.78–26.62 | 29.5±2.5 |
| MCHC (g/dL) <31.5 | 0 | 0 | | 33±1.5 |
| RDW <11.6 (fL) | 20 | 11.30±0.2038 | 11.20–11.39 | 12.8±1.2 |
| Platelets<150 (mm$^3$) | 4 (2.5) | 133.3±7.274 | 121.7–144.8 | 150–400 |
| Leukocytes <4.0 (mm$^3$) | 3 (1.8) | 133.3±7.274 | 2.515–4.471 | 7.0±3.0 |
| Neutrophils <2.0 (mm$^3$) | 0 | 0 | | 2.0–7.0 |
| Hepatic enzymes | | | | |
| AST (IU) | | | | |
| Men >50 | 2 (4.4) | 85.50±40.31 | 0 | 10–50 |
| Women >35 | 8 (7.0) | 50.63±6.968 | 44.80–56.45 | 7–35 |
| ALT (IU) | | | | |
| Men>50 | 3 (6.6) | 59.33±6.807 | 42.42–76.24 | 10–50 U/L |
| Women >35 | 12 (10.5) | 56.50±20.87 | 43.24–69.76 | 10–35 U/L |

ALT, alanine aminotransferase; AST, aspartate aminotransferase; MCH, mean corpuscular hemoglobin; MCHC, mean corpuscular hemoglobin concentration; MCV, mean corpuscular volume; NA, not available; RDW, red cell distribution width; NR, normal range.

In relation to the epidemiological characteristics of the study population, the mean age was 43.50±16.83 years (95% CI, 40.87–46.14; range, 18–89 years). The mean residence time in the city was 23.86±15.27 years (95% CI, 21.46–26.25; range, 1–68 years). There were a greater number of women 115 of 158 (72.3%) than men because there are more women in the area where the research was carried. The age-related distribution was as follows: >18–20 years, 21 (13.2%); 21–40 years, 52 (32.7%); 41–60 years, 55 (34.6%); and 61–80 years, 31 (19.5%).

## Discussion

In 2005, HVL reached the municipalities in RNHA11 mesoregion, western São Paulo, and in 2010 the municipalities of Pontal of Paranapanema, a poor and asymmetric region. In Teodoro Sampaio, considered the capital of Pontal of Paranapanema, *Lu. longipalpis* and CVL were found in 2010 and HVL in 2018. In 2018, 19 of 45 (42.2%) municipalities in RNHA11 recorded cases. A complex array of socioeconomic and environmental factors may be fueling the epidemic and sustaining endemic transmission of VL in this region (Fig 7).

The first *Lu. longipalpis* outbreak was found in the microregion of Dracena in 2003 [19]. One year later, vectors were found in Tupi Paulista and other adjacent municipalities, which suggested the existence of a possible route of dispersion by irradiation or by contiguity. However, *Lu. longipalpis* was registered in the region far from the epicenter, suggesting dispersion by jumps and toward the municipalities of the south, and HVL cases reached the

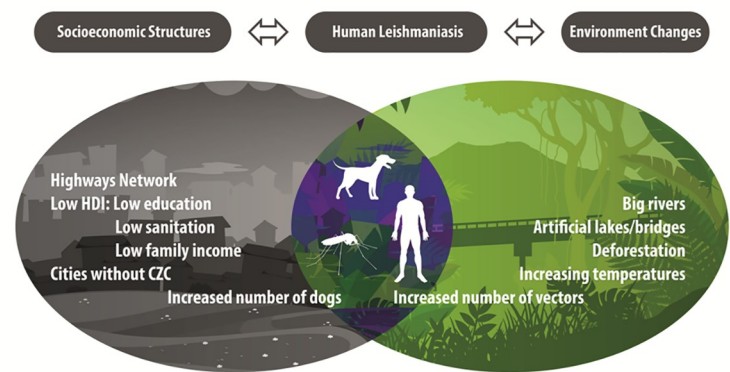

**Fig 7. Risk factors related to the transmission of visceral leishmaniasis in São Paulo State, Brazil.** Credit for the image's creation: Fernanda Lussari.

municipalities of Pontal of Paranapanema in 2010 (Fig 4A and 4B). An extensive highway network (2382 km), linking endemic regions to small- and middle-sized cities that may function as potential routes of dispersion of vectors and infected dogs was demonstrated in RNHA11 mesoregion. We suggest that the main axis adding to VL dispersion is the Euclides Figueiredo highway (SP562). Starting in Ilha Solteira outside RNHA11, the highway goes down toward the cities of Andradina and Dracena and toward the municipalities of Pontal of Paranapanema, crossing Teodoro Sampaio at the border of Paraná state. Nevertheless, dissemination may also occur in lower rates by a secondary axis, SPV031 and SPV035, starting near Brasilândia, an endemic region in MS, and following the Paraná River through small-sized cities and ending in Teodoro Sampaio must also be considered. All the municipalities crossed by these highways became endemic areas of VL (Fig 4A). The potential role of highways in the spread of VL in the southeastern and southern states of Brazil has been pinpointed in recent years [5,6]. However, a new route has been described and has gained great importance. Connecting Foz do Iguaçu in the west, on the Brazilian side of the triple border and considered endemic for VL, to the port of Paranaguá in the extreme east, a large number of trucks carrying grain, fertilizers, fuel, and goods, circulate daily on BR 277, the most important highway in Paraná state. It was suggested that following BR-277 and transects, coming from Argentina and Paraguay, seropositive dogs and *Lu. longipalpis* vector spread to the neighboring cities, becoming a region endemic to VL. Furthermore, potentially it is likely an important route of dispersion of infected dogs, vectors, and parasites to other regions of Paraná state [20,21].

In the western region, the proliferation of vectors could be aggravated by the extensive watershed flowing into the Paraná and Paranapanema Rivers, supporting hydroelectric dams, bridges and artificial lakes [22–26]. Porto Primavera dam in Paraná River has an artificial lake with a flooded area of 2.250 $km^2$ and divides the municipalities of RNHA11mesoregion and MS. Dividing São Paulo and Paraná state, the Paranapanema River flows on the border of ten municipalities with increasing rates of VL to the VL-free areas of Paraná state. Artificial lakes modify the climate of their surrounding areas as well as the environment within which the lake interacts [22–26]. In Presidente Epitacio, a city bordering the lake of Primavera dam, higher yearly temperatures rates of 23.93±3.06˚C (95% CI 22.54–25.32) were found compared with 22˚C in municipalities far from the lakes [23]. The influence of artificial lakes of hydroelectric dams in Paranapanema River on the ecological aspects of the sand fly fauna in 3 different settings of an endemic area of CL in Paraná state was investigated. A wide dispersion of sand flies involved in the transmission of leishmaniasis was found in all the settings [22]. In Panorama, a

city located on the border of the lake of the Paraná River, also an endemic area for VL, the high abundance of *Lu. longipalpis* had a significant correlation with temperature and humidity [27]. A relationship with climate and environmental factors with increased incidence rates of VL and annual precipitation, humidity, enhanced vegetation index, and night temperature values were found in the State of Tocantins, located in the northern region of Brazil [28].

In Teodoro Sampaio, although an entomological survey started in 2000, *Lu. longipalpis* sand flies were found only in 2010. It is well known that other phlebotomines, such as *Lu. migonei*, may be permissive vectors of VL in areas where *Lu. longipalpis* has not been recorded, but human or canine cases do occur [29–31]. According to the SUCEN database, in the period from 2000 to 2010, *Lu. cruzi* a VL vector found in Central-West of Brazil [32], and *Lu. migonei* were not identified in Teodoro Sampaio urban area. *Lu. migonei* was found in the wild forest of the Devil's Hill in research carried out for an LT investigation in 1994, 1995, and 1996.

Although a direct association between HDI and the presence of VL in municipalities of RNHA11 mesoregion was not found, linear regression showed a tendency of a lower number of infected individuals in municipalities with higher HDI. It is well known that poverty induces health care inequalities, poor housing conditions, lower income, less education, or lower occupational skill levels, and the population tend to be less healthy than those who experience higher levels of social conditions in other areas [16,32,33]. Supporting the data, demographic, environmental, and behavioral characteristics of inhabitants of rural settlements, a vulnerable population, have been determined. In Teodoro Sampaio, only 7.1% had a high school education, 11.9% lacked elementary instruction, and 77.3% had completed primary school. The income varied from less than 100 US$ (91.6%) to about 170 US$ (8.4%) per month. Similar rates were found in a neighboring rural settlement in Mirante of Paranapanema [34,35]. Globally and countrywide, VL is considered a neglected disease that affects the poorest of the poor [17,33,34].

In Teodoro Sampaio, on a local scale, the periphery of sector 1 had the higher number of infected dogs, demonstrated by critical point hotspots. The first human case of VL in the city was diagnosed in this sector. From 2010 to 2018, 298 (7.9%) dogs were found to be positive for CVL and 110 (36.9%) were euthanized. They were first found in sectors 1, 2, and 3 on the periphery, and in subsequent years, the hot spots spread to all urban areas. In the city, low levels of infected dogs were euthanized (36.9%). Similar to other cities of Pontal of Paranapanema, Teodoro Sampaio is surrounded by a high number of rural settlements [35,36]. Throughout the region, it is anecdotal that owners of dogs infected with VL hide their animals in these settings and may be disseminating the disease to rural areas. Although controversial, euthanasia is one of the most important actions supported by Brazilian Visceral Leishmaniasis Surveillance and Control Program in reducing the prevalence of *Leishmania (Leishmania) infantum chagasi* infection countrywide [37]. It has been shown that human epidemics of VL are usually preceded by or concomitant with CVL [10]. However, in Teodoro Sampaio, a diagnosis lag of 8 years between finding vectors, CVL, and HVL was found. Up to December 2020, only one individual had been registered. One possible explanation is that the city is located in the neighborhood of the Devil's Hill State Park, an endemic region for cutaneous leishmaniasis (CL) vectors [38,39], and the municipality is considered endemic for CL. There is some evidence that vectors have been present in the urban area for many years and a superposition CL/VL is occurring and a cross-reaction may be on course, protecting people from VL infection [40]. Supporting these findings, a VL serological survey of 159 individuals living in the urban area was carried out and only 2 (1.26%) individuals were positive by ELISA, with a probable false positive and/or cross-reaction. In addition, the mean residence times in the city was 23.86 ±15.27 years, increasing the possibility of these individuals being immunized against CL. Although VL-infected individuals do not have hallmark signs and symptoms, severe

pancytopenia (thrombocytopenia, leukopenia, and anemia) and increased levels of hepatic enzymes may be considered in the differential diagnosis in VL endemic areas [41]. In the population analyzed in this study, no laboratory findings symptomatic of chronic VL were found.

Shedding new light on the complex interactions occurring within the spread of VL in the western region and identifying the existence of spatial clusters of municipalities with increased vulnerability, Moran's I and LISA significance maps were created. Moran's I map identified a cluster of 13 municipalities with significant autocorrelation with CVL, and 5 municipalities were classified as high/high: Dracena, Ouro Verde, Tupi Paulista, Junqueirópolis, and Presidente Venceslau. Overall, 4 of these municipalities are considered the epicenter of the disease in which HVL was found in Dracena and Ouro Verde in 2005; Tupi Paulista in 2006, and Junqueirópolis in 2007 (Fig 4B). Furthermore, Dracena has a higher number of cases of HVL, varying from 58 to 176 infected individuals, followed by Ouro Verde, Tupi Paulista, and Junqueirópolis, varying from 17 to 57 infected individuals (Fig 4A). After all these years, despite the efforts made by the public authorities, Dracena remains the main focus of VL in western São Paulo. Taken together, these areas should be considered the target of local public policies and receive priority in surveillance actions within the scope of VL. As far as we know, this is the first effort to identify priority areas for VL in São Paulo State using Moran's I and LISA spatial analysis. In Brazil and worldwide, these tools have been used to identify vulnerable areas of parasitic, vector-borne neglected tropical diseases [14–16,42]. Moran's I and LISA techniques identified 27 priority areas for surveillance and control of VL in Belo Horizonte, an endemic area of Minas Geraes state [13]. In northeast municipalities, a VL Brazilian endemic region, spatial and space-time clusters of VL were identified in sertão and middle-north subregions, overlapping with high social vulnerability areas [14]. In Morocco, significant LISA clustering maps of human leishmaniasis have been identified in different provinces [42].

Different shortcomings should be considered in this study. The mean age of the population screened for VL was 43.50 years, and 72.33% were women. It is well known that VL affects mainly children and old people [40]. Thus, the population analyzed is not well representative of the disease. Municipal surveillance control programs of CVL were discontinued and there were difficulties in obtaining current records with a more accurate surveillance and response system to apply geospatial methods.

## Conclusion

In RNHA11 mesoregion, in a regional scale VL is spreading with a progressive extension, from the north, the epicenter, to the south, from endemic to non-endemic areas. A cluster of municipalities in the epicenter showed a significant autocorrelation for CVL and should receive priority in surveillance actions within the scope of VL. Socioeconomic and environmental factors may be fueling the epidemic and sustaining the endemic transmission of the disease (Fig 7). These findings further highlight the asymmetry of the western region compared with other regions of São Paulo state and add new knowledge on the spread of VL, contributing to the study of a neglected disease in a region of São Paulo, Brazil.

## Supporting information

**S1 Data. Supporting data.**
(XLSX)

## Author Contributions

**Conceptualization:** Luiz Euribel Prestes-Carneiro.

**Data curation:** Luiz Euribel Prestes-Carneiro.

**Formal analysis:** Elivelton Silva Fonseca, Lourdes Aparecida Zampieri D'Andrea.

**Funding acquisition:** Elivelton Silva Fonseca.

**Investigation:** Regiane Soares Santana, Karina Briguenti Souza, Cristiane Oliveira Andrade, Lourdes Aparecida Zampieri D'Andrea, Ivete Rocha Anjolete.

**Methodology:** Elivelton Silva Fonseca, Marcia Mitiko Kaihara Meidas, Lourdes Aparecida Zampieri D'Andrea, Francisco Assis Silva, Edilson Ferreira Flores, Ivete Rocha Anjolete.

**Project administration:** Luiz Euribel Prestes-Carneiro.

**Supervision:** Regiane Soares Santana.

**Validation:** Luiz Euribel Prestes-Carneiro.

**Visualization:** Fernanda Lussari, Elivelton Silva Fonseca, Francisco Assis Silva, Edilson Ferreira Flores.

**Writing – original draft:** Luiz Euribel Prestes-Carneiro.

**Writing – review & editing:** Regiane Soares Santana, Karina Briguenti Souza, Fernanda Lussari, Elivelton Silva Fonseca, Cristiane Oliveira Andrade, Marcia Mitiko Kaihara Meidas, Lourdes Aparecida Zampieri D'Andrea, Francisco Assis Silva, Edilson Ferreira Flores, Ivete Rocha Anjolete, Luiz Euribel Prestes-Carneiro.

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
