## [Decision Letter · Decision Letter 0]

29 Dec 2020

Dear Dr Prestes-Carneiro,

Thank you very much for submitting your manuscript "The burden of visceral leishmaniasis in western São Paulo: a neglected disease in this region of Brazil" for consideration at PLOS Neglected Tropical Diseases. As with all papers reviewed by the journal, your manuscript was reviewed by members of the editorial board and by several independent reviewers. In light of the reviews (below this email), we would like to invite the resubmission of a significantly-revised version that takes into account the reviewers' comments. 

The reviewers consider the topic and study population of interest. However, the reviewers request substantial revisions to clarify the study objectives, methods, and results. Specifically, please clearly state the hypothesis being addressed (reviewer 1), provide requested necessary details regarding different sources of the data including methodologies used by individual databases and how the those differences are reconciled in the analysis (reviewers 1 and 3).

We cannot make any decision about publication until we have seen the revised manuscript and your response to the reviewers' comments. Your revised manuscript is also likely to be sent to reviewers for further evaluation.

Sincerely,

Luther A Bartelt

Guest Editor

Nadira Karunaweera

Deputy Editor

The reviewers consider the topic and study population of interest. However, the reviewers request substantial revisions to clarify the study objectives, methods, and results. Specifically, please clearly state the hypothesis being addressed (reviewer 1), provide requested necessary details regarding different sources of the data including methodologies used by individual databases and how the those differences are reconciled in the analysis (reviewers 1 and 3).

Reviewer's Responses to Questions

**Key Review Criteria Required for Acceptance?**

**Methods**

-Are the objectives of the study clearly articulated with a clear testable hypothesis stated?

-Is the study design appropriate to address the stated objectives?

-Is the population clearly described and appropriate for the hypothesis being tested?

-Is the sample size sufficient to ensure adequate power to address the hypothesis being tested?

-Were correct statistical analysis used to support conclusions?

-Are there concerns about ethical or regulatory requirements being met?

Reviewer #1: The manuscript "The burden of visceral leishmaniasis in western São Paulo: a neglected disease in this region of Brazil " aims to determine the LV burden in 29 municipalities of the Regional Health Assistance Network (RNHA11), west of São Paulo, in a multiscale approach centered on the city of Teodoro Sampaio. However, the authors do not propose any hypothesis in the introduction section. They also do not raise questions about the entry and spread of the disease in the region. They only state that "In epidemiological studies, geospatial analysis has been used successfully for vector-borne diseases as a tool for space-time analysis and to predict the influence of visceral leishmaniasis in endemic areas of western São Paulo on the spread of the disease to other endemic areas. 

The study project presented may be appropriate to meet the stated objectives. The population is clearly described. 

However, the authors do not raise any hypothesis in the introduction. The data were obtained from public agencies where are obligatorily recorded. A small sampling of human, entomological and dog cases were performed as research; more specifically in Theodoro Sampaio. However, in the description it is necessary to be more clear which data were obtained from databases and which were performed by the authors. The ethical requirements were respected.

Reviewer #2: The objectives of the study are clearly articulated. The study desing is a little bit confuse for the reader. The population studied (human, vectors and dogs) are described in a disorderly way, in my opinion. It becomes a bit difficult to read the paper. The manuscript contributes with original and useful information related to several aspects of leishmaniasis in a poor area.

Reviewer #3: Are the objectives of the study clearly articulated with a clear testable hypothesis stated?

No, the authors used data from other public bodies as part of the methodology. For example: Data from the Supervision in Control of Epidemic- (SUCEN), Center of Regional Laboratory of the Adolfo Lutz Institute of Presidente Prudente V (CRL-CBSALI-PPV), National System of Diseases Notification (SINAN), Epidemiological Surveillance Center (CVE-SP). Considering each institution, the methodology used by them to obtain their results is different. If you need to redo this search. It will be necessary to obtain all this data and more that they have and if it is possible to do so.

Is the study design appropriate to address the stated objectives?

No, because results from other studies are used to complete with the results of the studied area, so to continue the analyzes

-Is the population clearly described and appropriate for the hypothesis being tested?

Yes

Is the sample size sufficient to ensure adequate power to address the hypothesis being tested?

Yes

Were correct statistical analysis used to support conclusions?

Yes

Are there concerns about ethical or regulatory requirements being met?

No

**Results**

-Does the analysis presented match the analysis plan?

-Are the results clearly and completely presented?

-Are the figures (Tables, Images) of sufficient quality for clarity?

Reviewer #1: In my opinion, the sections "Results" and "Materials and Methods" need to be improved. The sequence of data and a title are not very didactic for the readers. They must be put in the same order. 

The figures from 1 to 5 are correct. However, the Figure 6 does not make much sense. It should be deleted or give more meaning to the readers.

In the reference section, many scientific words are not italicized.

Reviewer #2: In my opinion, some part of the text that appear on the resuts must be moved to Material and Methods.

Reviewer #3: Does the analysis presented match the analysis plan?

Yes

-Are the results clearly and completely presented?

No, there was no entomological research, but a collection of sandflies, the results being incomplete. In the topic that talks about the dispersion route of visceral Leishmaniasis, a description of the geography and highways of the studied site were made, with public information, already known. This information would be better used if it were placed in the “Material and Methods”

-Are the figures (Tables, Images) of sufficient quality for clarity?

The exception of table 1. The other tables and images are of sufficient quality for clarity

**Conclusions**

-Are the conclusions supported by the data presented?

-Are the limitations of analysis clearly described?

-Do the authors discuss how these data can be helpful to advance our understanding of the topic under study?

-Is public health relevance addressed?

Reviewer #1: The manuscript is of interest to public health. However, the authors should improve the discussion section. The authors should be more concise and didactic.

Reviewer #2: Part of the discussion and conclusions must be supported by more references.

Reviewer #3: Are the conclusions supported by the data presented?

Yes

Are the limitations of analysis clearly described?

Yes

Do the authors discuss how these data can be helpful to advance our understanding of the topic under study?

No, they report that this study may contribute new knowledge to the study of this neglected disease. They cite environmental, social and economic conditions as factors that can hinder actions against CVL and disease vectors. Thus sustaining this disease in an endemic area.

-Is public health relevance addressed?

Yes

**Editorial and Data Presentation Modifications?**

Reviewer #1: Authors should place the hypotheses in the introductory section, present the results in a didactic manner and shorten the discussion. They should focus only on what they found, without making disconnected sentences and statements without referenes citations

Reviewer #2: (No Response)

Reviewer #3: (No Response)

**Summary and General Comments**

Reviewer #1: Authors should place the hypotheses in the introductory section, present the results in a didactic manner and shorten the discussion. They should focus only on what they found, without making disconnected sentences and statements without referenes citations

Reviewer #2: (No Response)

Reviewer #3: Dear Authors,

The article is good and relevant. However, the summary should be clearer, data more accurate and transparent. The statistical calculations used should be mentioned, how this study will contribute to the practical actions of controlling and preventing the disease, vector and treatment of the host. In the methodology, check the information made in the summary about the entomological research data and unify the information. In the results, inform only the data obtained when conducting the research. Make a Discussion considering only other works on visceral leishmaniasis, the vectors, the etiological agent and their mammalian hosts.

PLOS authors have the option to publish the peer review history of their article (what does this mean?). If published, this will include your full peer review and any attached files.

Reviewer #1: No

Reviewer #2: No

Reviewer #3: No

Figure Files:

Data Requirements:

Reproducibility:

To enhance the reproducibility of your results, PLOS recommends that you deposit laboratory protocols in protocols.io, where a protocol can be assigned its own identifier (DOI) such that it can be cited independently in the future. For instructions see https://journals.plos.org/plosntds/s/submission-guidelines#loc-methods.

---

## [Decision Letter · Decision Letter 1]

17 Apr 2021

Dear Dr Prestes-Carneiro,

Thank you very much for submitting your manuscript "Cases and distribution of visceral leishmaniasis in western São Paulo: a neglected disease in this region of Brazil" for consideration at PLOS Neglected Tropical Diseases. As with all papers reviewed by the journal, your manuscript was reviewed by members of the editorial board and by several independent reviewers. The reviewers appreciated the attention to an important topic. Based on the reviews, we are likely to accept this manuscript for publication, providing that you modify the manuscript according to the review recommendations. 

Sincerely,

Luther A Bartelt

Associate Editor

Nadira Karunaweera

Deputy Editor

Reviewer's Responses to Questions

**Key Review Criteria Required for Acceptance?**

**Methods**

-Are the objectives of the study clearly articulated with a clear testable hypothesis stated?

-Is the study design appropriate to address the stated objectives?

-Is the population clearly described and appropriate for the hypothesis being tested?

-Is the sample size sufficient to ensure adequate power to address the hypothesis being tested?

-Were correct statistical analysis used to support conclusions?

-Are there concerns about ethical or regulatory requirements being met?

Reviewer #1: Now the methods are well presented, correct statistics have been used. The corrections made have certainly improved the manuscript.

Reviewer #3: The hypotheses defended by the authors were presented in a more detailed and objective way. The target population is described so that the methodology used can be understood, thus allowing it to be tested later. The sampling used was sufficient to carry out the suggested statistical analysis, which contributed to the understanding of the research conclusion. Ethical procedures have been described satisfactorily and in accordance with standards.

**Results**

-Does the analysis presented match the analysis plan?

-Are the results clearly and completely presented?

-Are the figures (Tables, Images) of sufficient quality for clarity?

Reviewer #1: Now the analysis presented match the analysis plan. The results are clearly and completely presented and finaly the figures (Tables, Images) have sufficient quality

Reviewer #3: The results were presented in a clear, detailed and objective manner. The figures are within the requested standards.

**Conclusions**

-Are the conclusions supported by the data presented?

-Are the limitations of analysis clearly described?

-Do the authors discuss how these data can be helpful to advance our understanding of the topic under study?

-Is public health relevance addressed?

Reviewer #1: Yes the manuscript has public health relevance adressed.

Reviewer #3: The conclusion is concise and clear, allowing an understanding of the study carried out. This work is relevant to public health, which is addressed in a satisfactory way in the conclusion.

**Editorial and Data Presentation Modifications?**

Reviewer #1: The manuscript can now be accepted

Reviewer #3: The authors followed the suggestions made, requiring only three modifications. Therefore, I recommend a "Minor Revision". If answered, accept it.

**Summary and General Comments**

Reviewer #1: The manuscript can now be accepted

Reviewer #3: To the Authors, I know it took some work to do the previous revision, but you did ... Congratulations. The article has improved a lot. I am recommending a “Minor Revision”, I ask you to pay special attention to the observation made in the discussion section of the attached document where I have made comments directly on your manuscript. If you want to publish the review, you can do so, no problem. I wish you success.

PLOS authors have the option to publish the peer review history of their article (what does this mean?). If published, this will include your full peer review and any attached files.

Reviewer #1: No

Reviewer #3: Yes: Dr. João Ricardo Carreira Alves

Figure Files:

Data Requirements:

Reproducibility:

To enhance the reproducibility of your results, we recommend that you deposit your laboratory protocols in protocols.io, where a protocol can be assigned its own identifier (DOI) such that it can be cited independently in the future. Additionally, PLOS ONE offers an option to publish peer-reviewed clinical study protocols. Read more information on sharing protocols at https://plos.org/protocols?utm_medium=editorial-email&utm_source=authorletters&utm_campaign=protocols.

References

---

## [Editor Report · Decision Letter 2]

26 Apr 2021

Dear Dr Prestes-Carneiro,

We are pleased to inform you that your manuscript 'Cases and distribution of visceral leishmaniasis in western São Paulo: a neglected disease in this region of Brazil' has been provisionally accepted for publication in PLOS Neglected Tropical Diseases.

Best regards,

Luther A Bartelt

Associate Editor

Nadira Karunaweera

Deputy Editor

---

## [Editor Report · Acceptance letter]

9 Jun 2021

Dear Dr Prestes-Carneiro,

We are delighted to inform you that your manuscript, "Cases and distribution of visceral leishmaniasis in western São Paulo: a neglected disease in this region of Brazil," has been formally accepted for publication in PLOS Neglected Tropical Diseases.

Best regards,

Shaden Kamhawi

co-Editor-in-Chief

Paul Brindley

co-Editor-in-Chief
